# RIPpore: A Novel Host-Derived Method for the Identification of Ricin Intoxication through Oxford Nanopore Direct RNA Sequencing

**DOI:** 10.3390/toxins14070470

**Published:** 2022-07-09

**Authors:** Yan Ryan, Abbie Harrison, Hannah Trivett, Catherine Hartley, Jonathan David, Graeme C. Clark, Julian A. Hiscox

**Affiliations:** 1Institute for Infection, Veterinary and Ecological Sciences, University of Liverpool, Liverpool L3 5RF, UK; yryan@liv.ac.uk (Y.R.); psaharri@liverpool.ac.uk (A.H.); h.trivett@liverpool.ac.uk (H.T.); catherine.hartley@liverpool.ac.uk (C.H.); 2Defence Science Technology Laboratory, Salisbury SP4 0JQ, UK; jdavid@mail.dstl.gov.uk

**Keywords:** ricin, RNA sequencing, MinIon, ribosome inactivating proteins, saporin, ribosomes

## Abstract

Ricin is a toxin which enters cells and depurinates an adenine base in the sarcin-ricin loop in the large ribosomal subunit, leading to the inhibition of protein translation and cell death. We postulated that this depurination event could be detected using Oxford Nanopore Technologies (ONT) direct RNA sequencing, detecting a change in charge in the ricin loop. In this study, A549 cells were exposed to ricin for 2–24 h in order to induce depurination. In addition, a novel software tool was developed termed RIPpore that could quantify the adenine modification of ribosomal RNA induced by ricin upon respiratory epithelial cells. We provided demonstrable evidence for the first time that this base change detected is specific to RIP activity using a neutralising antibody against ricin. We believe this represents the first detection of depurination in RNA achieved using ONT sequencers. Collectively, this work highlights the potential for ONT and direct RNA sequencing to detect and quantify depurination events caused by ribosome-inactivating proteins such as ricin. RIPpore could have utility in the evaluation of new treatments and/or in the diagnosis of exposure to ricin.

## 1. Introduction

Depurination is where either an adenine or guanine base is enzymatically or chemically cleaved from the phosphodiester backbone, leaving a gap in the DNA or RNA strand. Ribosome-inactivating proteins (RIPs) are a class of toxin which inhibits protein synthesis via the depurination of a specific adenine base present within a conserved ribosomal region known as the sarcin-ricin loop (SRL) [1]. RIPs induce irreversible hydrolysis of the N-glycosidic bond at adenine 4605 (A4605) in *Homo sapien* 28s rRNA [2]. Therefore, A4605 depurination is an ideal candidate for the potential identification of RIP exposure as this results in the modification of a single known target site. 

Ricin is one of the most well-known RIPs due to potent cytotoxic properties and is a significant risk to both human and animal health, with many deliberate exposures having been reported, including the assassination of Georgi Markov, a Bulgarian dissident writer [3,4]. Ricin is a plant lectin and holotoxin composed of two subunits individually referred to as ricin toxin A-chain (RTA) and ricin toxin B-chain (RTB). Ricin B-chain binds to cell surface galactosides and facilitates entry, whilst the A-chain catalyses depurination, classifying ricin as a type II RIP. Related type I RIPs, such as saporin, lack this B-chain and, therefore typically, require the use of an artificial cell entry mechanism (i.e., transfection reagent or antibody conjugation) to enter a cell and cause toxicity. Saporin is dependent on cell type, enter cells via endocytosis of uncoated vesicles [5], or bind to the macroglobulin receptor α2 [5,6].

Depurination spontaneously occurs in DNA at a rate of 2000–10,000 events per day per cell [7], which are broadly resolved by base excision repair [8]. However, mis-repairing can occur, which leads to mutation and, in some cases, oncogenesis [9]. The depurination of RNA, depending on context, has multiple downstream effects such as halting translation [10], preventing the packaging of viral RNA into capsids [11], or inhibiting protein synthesis in ribosomes by cleavage of a key catalytic adenine [12]. Oxford Nanopore Technologies sequencing can potentially capture the entire spectrum of modifications present on nonamplified DNA or RNA. The system works by measuring the ionic current and “dwell times” of nucleotides traversing through a nanopore. Basecalling software (basecallers) takes a signal representing a nucleotide or sequence of nucleotides from a sequencing device and translates this into the constituent nucleotides in a format that can be analysed. ONT has developed basecallers for Nanopore sequencing (e.g., Guppy) that translate ionic current and dwell time into the nucleotides AGCT/U. In contrast to other sequencing methodologies, the nucleic acid being sequenced on an ONT flow cell is analysed across the nanopore, five nucleotides at a time (i.e., denoted as a kmer). Newer versions of Guppy can infer 6mA (N6 methyladenine) and 5mC (5-methylcytosine) in DNA. Currently, no ONT basecallers have yet been trained to recognise modifications on RNA, and downstream detection tools rely on comparisons between unmodified and the knockout or knock downs of RNA-modifying proteins to identify specific modifications. 

To investigate whether depurination of A4605 in the SRL could be identified by direct RNA sequencing, a lung epithelial cell line (A549) was exposed to ricin. Post-exposure, direct RNAseq was used to evaluate the ricin-induced signal alterations in 28s rRNA. The data indicated that depurination of A4605 formed a signal analogous to what would occur if the base were U4605 due to the activity of ricin. We describe the development of RIPpore, a Python script which leveraged the transposition to quantify and isolate depurinated reads, enabling signal-level investigation of wholly modified and unmodified populations. The methodology represents a novel use of ONT dRNA sequencing to identify SRL depurination in a sample exposed to RIPs. The depurination measured is specific to RIP activity through the use of a neutralising antibody to ricin. Collectively, this work opens a door to the future transcriptome or genome-wide mapping of apurunic sites and the detection of RIP toxins through ONT direct RNA sequencing. 

## 2. Results

In order to identify depurination, a combined biological exposure, sequence, and computational strategy needed to be developed. Cultured A549 cells were exposed to ricin or saporin (two RIPs) and the resulting RNAs were sequenced, by direct RNAseq using ONT flow cells (Figure 1).

### 2.1. Ricin Purification and Cytotoxicity Assay in Cell Culture

The ricin toxin used in this study was isolated from the *Ricinus communis* (*subsp. Zanzibariensis*) plant and underwent a two-stage purification process from bean, to crude, to purified preparation. The purification of ricin was confirmed by Coomassie blue staining, with the solvent-extracted crude preparation showing multiple products, and a single product of the expected molecular weight after HPLC purification (Appendix A). To assess the toxicity of the crude and purified ricin, a WST-1 cytotoxicity assay was performed on the cell lines of choice A549 cells (Appendix A). (Note that the complete Commassie blue staining pattern is shown in Appendix A).

Cell viability plateaued at concentrations of ricin above 10 pM, and further increases in concentration did not decrease cell viability, suggesting that these were saturating amounts. The half maximal effective dose (EC50) had only a 2 fM difference between preparations. To ensure saturation and to provide parity to other methodologies [13], a 1 nM concentration of purified ricin was used throughout this study. Purified ricin toxin was also used to reduce the possibility that effects were not due to other constituents of castor beans.

### 2.2. 28s Ribosomal RNA-Targeted Oligo Design

To directly sequence the depurinated base(s) in the SRL, ONT Reverse Transcription Adapter (RTA) Oligonucleotides A and B (designed to ligate to polyA-tailed mRNAs) were customised to specifically ligate to nonpolyadenylated ribosomal RNA. The 3′ polyT run was replaced with the reverse complement of the final 10 3′ bases of the 28s rRNA (producing: GACAAACCCT). For flow cell efficiency, barcoding was used to multiplex samples. Appendix A describes the number of reads and samples per flow cell; Appendix A shows an overview of the per sample read count, the coverage of position 4605 in the 28s rRNA, and the percentage of reads which mapped to the 28s rRNA. Once ligated, samples were reverse-transcribed, generating a cDNA hybrid for enhanced stability and to minimise secondary structures in the 28s RNA, facilitating the transit of the native RNA molecule through the nanopore for sequencing. 

### 2.3. Development of RIPpore: A Tool for Measuring Depurination

Direct RNAseq using Oxford Nanopore enabled the study of chemical modifications on RNA by measuring potential alterations in charge. The hypothesis was tested where the depuration of adenine 4605 by an RIP would lead to a detectable signal change or miscalling of bases. To test this hypothesis, hACE2-A549 cells were exposed to ricin or saporin (a type I RIP), and the total RNA was extracted and sequenced directly. Reverse transcription was performed to aid in the sequencing of the 28s Guppy to call the sequence of the 28s rRNA moving through the pore. We predicted that if the apurinic 4605 site was present, this would be potentially miscalled as a base, whereas an unmodified base would be correctly called as an adenine.

Direct sequence reads were mapped to human 28s rRNA using minimap2 (although uracil was basecalled as thymine, this is referred to as an uracil residue). At 2 h post-ricin exposure, 12% of nucleotides at the site of depurination (A4605) had an apparent change from adenine to uracil compared to the equivalent sequence from unexposed cells (Table 1). 

To quantify and further study this in the multiple samples, a python script was developed, RIPpore, to count the type and number of each nucleotide occurring at base 4605. RIPpore functions by taking a sam file (with reads aligned to the 28s rRNA) and using *pysam* to take the pileup column at position 4605. Subsequently, RIPpore calculates the proportion of thymine bases, using the percentage of thymine as a proxy for depurinated reads. By analysing the unexposed controls and cells exposed to ricin B-chain (RTB), a baseline level of noise was established with ~0.3% of reads called as uracil (Table 1). This identified a limit of detection of the approach, whereby RIP-induced uracil transpositions above a cut-off of 0.4% were reported as depurinated. This cut-off was calculated by taking the maximum percentage plus standard deviation of uracil bases called at position A4605 in control and RTB exposed cells. Updating Guppy from version 3.0.6 to 4.4.2 and repeating basecalling on control data led to a decrease in the noise of uracil from ~1.5% (data not shown) to a mean of 0.277 (Table 1). Re-basecalling data from RIP-exposed cells found no change in uracil calls (data not shown). An increase in apparent depurination events was also observed in 28s rRNA purified from cells exposed to saporin.

Boxplots were used to visualise counts of each nucleotide at base 4605 across all samples. The data showed a time-dependent increase in depurination, by the concomitant increase in uracil and decrease in adenine (with significance determined by pairwise *t*-tests) (Figure 2). The proportion of cytosine and guanine both increased in count to around 3% each at this site. The depurination effect of ricin was detected at the initial time point analysed (2 h post-exposure), peaked at 6 h, and had declined by 24 h. All ricin exposures showed a significant difference at this position from the negative control or RTB. Incorrect basecalling of cytosine and guanidine did occur, but only to a maximum of 3.5% of reads, with guanidine increasing by 1% over the control (statistically significant at later time points). Cytosine increased between 1 and 3%, with a large sample variability for the same conditions, and only saporin exposure showed a significant difference.

The change in depurination levels by the concentration of ricin is not linear, with a 10× reduction in ricin from 1000–100 pm resulting in a level of depurination of 30–2.1%, appearing to become asymptotic with further reductions having a lesser decrease in levels of depurination (Figure 3). 

Use of the ovine antitoxin shows a reduction in depurination (Figure 4) to a level below that induced by 1 pm of exposure (Table 1), with an average of 0.48% of calls being uracil compared to 1 pm having an average of 0.55%. It is noteworthy that although there was reduced depurination in the antitoxin-treated ricin-exposed cells relative to the control, this indicates partial rather than complete neutralisation of toxin activity by the antibody. However, this partial neutralisation is the equivalent of an over 1000-times reduction in concentration of ricin exposure.

### 2.4. Charge Intensity Analysis of the Ricin Loop Shows a Shift Caused by Depurination

In order to further investigate the miscalling event assigned by guppy, we sought to study the perturbations in signal at the sarcin-ricin loop. As the miscalling of apurinic 4605 as uracil appears consistently dose-dependent and responded as expected to controls and antitoxin-based neutralisation, we examined how the charge of depurinated A4605 compares to U4605. As the variations in raw measurements such as dwell time and charge intensity are utilised by basecalling algorithms to predict the nucleotide sequence, the influence that depurination in the 28s rRNA had upon these raw features was measured. A previous study suggested that the measurement of transition dynamics of depurinated DNA showed an increase in dwell time using solid-state nanopores [14] (NB: which has no relation to ONT and is not a sequencing device). To do so, charge intensities of the ricin loop were simulated using Nanocompore such that an A4605 was compared to an U4605 (Figure 5A,B, respectively). A sample from ricin-exposed cells would contain sequence reads that are either called A4605 or U4605. These were matched, using Tombo [15], to the actual recorded charge intensities in the ricin loop from the control (predominately A4605) or from cells exposed to ricin (Figure 5C,D, respectively). Tombo was used to superimpose the recorded charge intensities onto the predicted charge intensities from the reference sequence. The sequence reads from cells exposed to ricin were segregated to only those called U4605, and the reads mapping to A4605 were disregarded (Figure 5D). The reference sequence was modified to U4605 and aligned to the actual charge intensity when the reads containing U4605 were considered in isolation.

The simulated reads showed a marked decrease in charge intensity at base 4605 comparing A4605 (Figure 5A) to U4605 (Figure 5B). 

This paralleled the findings from the Nanocompore analysis (Figure 5B) which showed a decrease in charge associated with an apurunic site at position 4605. See Appendix A for simulations with cytosine and guanidine at position 4605. This suggested that Guppy was miscalling the apurunic site as an uracil. When viewing the charge intensities through Tombo (Figure 5C), the control reads aligned closely with the predicted charge for A4605, whereas, the immediate nucleotide upstream of site 4605 decreased in charge intensity with depurination of site 4605. This was observed for both the actual and predicted charge intensities. This identified the likely mechanism for Guppy basecalling the apurunic site 4605 and supported the use of quantifying uracil calls as a proxy for depurination.

To further investigate these signal-level changes, Nanocompore was used to study charge intensity and dwell time at a per base resolution. The changes induced by depurination were compared between the different exposures and controls and were visualised using Nanocompore, which curiously shows the depurinated adenine as base 4602, a difference of three nucleotides compared to its position in the reference; due to this, it is referred to as 4602 in the context of Nanocompore and displayed as such in the figures. The data indicated that there was a decrease in charge associated with the apurunic site, with the proportion of reads with this shift increasing with time, in RNA from cells exposed to either ricin or saporin when compared to the unexposed controls (Figure 6).

Nanocompore identified a distinct increase in charge density formed below 107 mean intensity units (Nanocompore unit), which was absent in the control reads and increases in a time-dependent manner. This effect was seen initially as a small peak in the exposed samples, which increased in size over time to a distinct peak, with a concomitant decrease in the canonical peak at 115 mean intensity units. Ricin exposure for 2, 4, 6, and 24 h (Figure 6A–D), respectively, caused a rise in reads at a lower charge density, increasing from 2 to 6 h, peaking, and plateauing between 6 h and 24. The 2 h exposure (Figure 6B) had the lowest significance in difference to the control of the ricin exposures (Figure 6G,H), with each subsequent exposure increasing in significance. Exposure with the ricin B-chain only had little discernible difference in charge density at position 4602, but a slight increase in dwell time (Figure 6E,G). Exposure of cells to saporin (Figure 6F) resulted in an increase in reads with a lower charge density and a slight increase in dwell time, which was greater than those observed in cells exposed to B-chain only. Ricin exposure produced a noticeable and significant increase in lower-charge reads at site 4602, in a time-dependent manner.

The effect not only occurred at site 4602 but altered both the charge and dwell time of nucleotides upstream and downstream, but no further than three nucleotides (Figure 5). An assessment of how many nucleotides up and downstream of position A4602 were affected by depurination was performed by extending the window used by Nanocompore to measure the charge density and dwell time (Figure 7).

B-chain produced little effect compared to controls (Figure 7A). A 6 h exposure to ricin resulted in the signal changes from the abasic site extending primarily across nucleotides 4600–4603 (Figure 7B). Nucleotide position 4603, as with position 4602, showed a decrease in charge intensity, whereas position 4600 showed an increase in charge intensity. There was little change in charge density at position 4601, but an increase in dwell time, indicating a slower translocation speed, corroborating the effect of depurination observed previously [14]. See Appendix A for violin plots of charge density and dwell time across all conditions.

## 3. Discussion

Here, we have demonstrated the first detection of depurination with direct RNA sequencing on an ONT flow cell (i.e., using either the MinION or GridION sequencer), by utilising the hyper-specific removal of adenine 4605 by ricin to only a single abasic site. This modification was identified through the development of direct sequencing primers specific to 28s ribosomes, and the basecalling of direct RNA sequence data producing the transposition of A4605 to U4605 in depurinated reads (i.e., the nucleotide modification induced the following ricin exposure; Figure 3). This was achieved through the development of a novel tool to quantify these events: RIPpore. To the authors knowledge, this is also the first detection of depurination in RNA achieved on ONT sequencers.

To investigate why the modified base was identified as uracil and how this could be exploited as a way of detecting depurination and RIP activity, we compared the signal level following exposure of cultured lung epithelial cells to ricin. The data indicated that the charge intensity and the dwell time were different at base 4605 from RNA obtained from cells exposed to ricin and saporin internalised via transfection reagents, compared to unexposed controls (Figure 4 and Figure 5). This provided a likely explanation for Guppy basecalling depurinated 4605 as uracil. To validate this, Nanocompore was used to simulate the 28s rRNA with U4605 and produced reads with a lower charge at the 4605 site versus A4605 (Figure 6A,B). Tombo produced a canonical model with expected charge intensities for a given sequence, which was compared against the recorded charge intensities. The analysis using Tombo, comparing U4605 and depurinated reads with A4605 and nondepurinated reads, concurred with the Nanocompore simulated reads (Figure 6C,D). The lower charge associated with depurination at 4605 was the cause of miscalled bases as U4605 and gave credence to the quantification of RIP-induced depurination by measuring levels of U4605 with RIPpore. Simulated data were used instead of the generation of ssRNA oligonucleotides containing U4605, as the synthesis process is limited to a maximum size of ~80 bp, whereas ONT sequencing has historically struggled with reads shorter than 150 bp. It was only recently that a software update has been made available that can accurately identify and basecall shorter read lengths to a minimum of 20 bp in length [16]. The issue of read lengths and the fact that bonito (i.e., the ONT tool used to train basecaller models for guppy) is not designed to work on bases outside of ATCGU make it difficult to train a basecaller that is able to detect depurination. In the future, if the synthesis or read length of oligos containing uracil at base 4605 can be managed, it may be theoretically possible to train a basecaller that can identify the depurination of base 4605 and that could differentiate an abasic site from U4605. This would require a tool to specifically replace U at depurinated bases in the basecalled data with an appropriate symbol (D for depurination or R for ribose backbone) as part of the training dataset and modification of Bonito (currently used by ONT to train basecalling models) to handle additional bases outside of ATCGU, which is not available at present or planned, to the best of our knowledge.

This methodology is proven with concentrations of ricin from 1000 to 1 picomole. As lethal exposures to ricin typically require upward of nanomolar to micromolar quantities in order to mediate its toxic effects [3,17], the picomole sensitivity of this technique means its use for the diagnosis of toxin exposure is worth further investigation. Currently, clinical diagnosis of ricin is challenging through signs and symptoms and/or the direct detection of the toxin using physical techniques (e.g., ELISA and mass spectrometry). RIPpore perhaps offers a novel indirect identification of ricin exposure through the identification of the specific depurination event induced during intoxication.

RIPpore allowed for quantification of RIP-induced depurination and demonstrated the time-dependent nature of the effect of ricin exposure on 28 s ribosomes. This matched the observation using digital droplet PCR (ddPCR) [13]. However, we believe our method may have more accurate results as it relies on mature 28s rRNA in order for the oligo to bind to the final 10 3′ bases, whereas PCR methodologies will likely also count any immature 28s rRNA transcripts that might be present. Our study found that after 24 h of exposure, the proportion of depurinated ribosomes had begun to decrease. Whether this effect was due to cell death removing depurinated ribosomes or the transcription of additional 28s rRNA to combat the loss of functional ribosomes is currently unclear. There is also the possibility that the cell was attempting to recover by creating additional ribosomal RNA, but was unable to translate the required proteins for correct ribosome folding which may then circumvent the action of ricin, resulting in increased nondepurinated reads [18].

For creating and testing countermeasures to RIPs, such as ricin, direct RNA sequencing will be highly useful in quantifying the relative protection against depurination/ricin exposure a treatment provides. RIPpore has also demonstrated that an antitoxin to ricin effectively prevents depurination to a level of an over 1000-times reduction in concentration of ricin exposure. RIPpore provides a quantitative assessment of depurination from direct sequence read data. This technique would also be of benefit to those developing RIP-based immunotoxins to assess off-target ribosome inactivation across various cell types and targets. This method also creates an additional wealth of data on the epi-state of the 28s rRNA due to other modifications, which, as of yet, is a relatively unexplored field using direct RNA sequencing. This will enhance investigation into ribosomal regulation due to differing modifications in a wide variety of contexts.

## 4. Materials and Methods

### 4.1. Ricin Extraction

The ricin preparations used for this investigation were conducted at Dstl. Ricin was extracted from castor beans of *Ricin communis subsp. zanzibariensis* as per the methodology previously published [19]. 

### 4.2. Assessment of Ricin Purification

Assessment of ricin preparations by SDS electrophoresis was carried out. The purity of the ricin preparations was assessed by SDS-PAGE using coomassie blue staining (GE Healthcare). Samples (1 mL at 5 or 10 mg/mL) were mixed with a 1:1 volume of sample buffer Tris-HCl (125 mM) pH to 6.8, SDS (10%, *w*/*v*), and glycerol (50%, *v*/*v*). The samples were heated at 95 °C for 5 min prior to loading 1 mL onto the gel (polyacrylamide gradient (4–14%); GE Healthcare). Proteins were separated for 40 min before Coomassie blue staining for visualisation, using the manufacturer’s instructions (Calibrated Densitometer GS-800; Bio-Rad, Hemel Hempstead, UK). Densitometric scans were analysed using Quantity One software (Bio-Rad). The purity of ricin was determined as a percentage of the total protein in each lane, using the average density analysis tool.

### 4.3. Cell Line

Cells of cell line hACE-A549 were kindly provided by Olivier Schwartz [20]. Cells were cultured in 10% foetal bovine serum and Dulbecco’s modified Eagle’s medium (DMEM) at 37 °C and 5% CO_2_. Cells were cultured to 80–90% confluency in a T75 before exposure. A549 (86012804) cells were obtained from the European Collection of Animal Cell Cultures (ECACC) (Public Health England, Salisbury, UK). Cells were maintained in culture medium consisting of DMEM (Sigma Aldrich, Poole, UK) with 10% (*v*/*v*) foetal calf serum (Sigma Aldrich, Poole, UK), 1% penicillin, streptomycin solution (containing 100 units/mL of penicillin and 0.01 mg/mL of streptomycin), and 1% (*w*/*v*) l-glutamine (Sigma Aldrich, Poole, UK) 2 mM.

### 4.4. In Vitro Cytotoxicity Assay

Cells were grown in T75 flasks in a humidified atmosphere of 5% CO_2_ in air at 37 °C and removed from the flask surface using incubation with trypsin (0.05% *w*/*v*) (Sigma Aldrich, Poole, UK) containing EDTA (0.03% *w*/*v*) (Sigma Aldrich, Poole, UK) on achieving 70% confluency. The monolayer was allowed to adhere to the culture plates for 24 h before addition of ricin. For toxicity assessment, ricin toxin was diluted to 100 ng mL^−1^ in culture medium and filtered using a 0.2 m sterile filter before further dilution in culture medium and addition to the assay plate in triplicate. The plates were then incubated for 48 h prior to the addition of 10 mL of Roche Cell Proliferation Reagent WST-1 (Sigma Aldrich, Poole, UK). After 3 h, the absorbance was read on a Thermo Multiskan plate reader (Thermo Fisher Scientific, Loughborough, UK) at 450 nm to assess cell viability.

### 4.5. Antitoxin-Based Ricin Neutralisation

An ovine anti-ricin antitoxin that has previously been demonstrated to have neutralising properties toward the toxin [21] was kindly provided by Lucy Cork (Dstl). Ricin (500 nm) and anti-toxin (50 mg/mL) were mixed (1:1; vol/vol) and then incubated for 30 min at room temperature. This ricin-antitoxin cocktail was diluted into DMEM (1 nm) before being administered to T75 flasks of A549 cells.

### 4.6. RIP Exposure

All RIPs were handled with appropriate safety measures. Ricin was added to cell culture media of a volume appropriate to the cell culture plate or flask being used to create a solution of 1 nM. Saporin was added as described by Rust et al., at 100 nM [22] for 24 h. B-chain was added at 1 nM for 6 h. Ricin exposure lasted 2, 4, 6, and 24 h. Cell culture media were then disposed of in sodium hypochlorite solution for inactivation. Care was taken not to generate aerosols at this step. Two PBS washes of the monolayer were performed to remove any remaining RIPs and culture media present. Waste PBS was disposed of in sodium hypochlorite solution. 

### 4.7. RIP Disposal

RIPs were deactivated by disposal into a solution of sodium hypochlorite with a minimum of 10,000 ppm active chlorine for 24 h. Sodium hypochlorite and inactivated RIPs were disposed in accordance with local regulations.

### 4.8. RNA Extraction

RNA was extracted using Trizol solution (Thermo Fisher) and optionally in conjunction with Phasemaker tubes (Thermo Fisher) according to the manufacturer’s instructions. 

### 4.9. Quantifying RNA Content

The RNA content of samples was measured using the Qubit RNA system (Thermo, Qubit RNA HS Assay Kit, Q32855). The Qubit working solution and standards were prepared as per the manufacturer’s instructions. An amount of 2 μL of each sample was then added to a tube containing 198 μL of working solution and was vortexed for 2–3 sections All tubes were incubated at room temperature for 2 min. Samples were read using a Qubit 3.0 Fluorometer using the high-sensitivity method. 

### 4.10. Targeted Direct RNA Oligos A and B with and without Barcodes

The following oligos were synthesised from Eurofins with HPLC purification, and were used to replace the standard Oligos A and B in the direct RNA sequencing protocol. The last 10 nucleotides of Oligo B are the reverse complement of the last 10 nucleotides 3′ of 28s rRNA AGGGTTTGTC resulting in GACAAACCCT from NCBI accession: NR_003287.4. Unbarcoded reads may modify ONT’s oligo A and B as described in Oxford Nanopore sequence-specific direct RNA protocol DSS_9081_v2_revK_14Aug2019. Deeplexicon and Poreplex [23,24] oligos were Standard Oligos modified to replace the poly T tail with the 28r rRNA sequence-specific sequence.Standard ONT oligosOligo A: 5′-/5PHOS/GGCTTCTTCTTGCTCTTAGGTAGTAGGTTC-3′Oligo B: 5′-GAGGCGAGCGGTCAATTTTCCTAAGAGCAAGAAGAAGCCGACAAACCCT-3′Deeplexicon 1OligoA1: 5′-/5Phos/GGCTTCTTCTTGCTCTTAGGTAGTAGGTTC-3′OligoB1: 5′-GAGGCGAGCGGTCAATTTTCCTAAGAGCAAGAAGAAGCCGACAAACCCT-3′Deeplexicon 3OligoA2: 5′-/5Phos/GTGATTCTCGTCTTTCTGCGTAGTAGGTTC-3′OligoB2: 5′-GAGGCGAGCGGTCAATTTTCGCAGAAAGACGAGAATCACGACAAACCCT-3′Deeplexicon 2OligoA3: 5′-/5Phos/GTACTTTTCTCTTTGCGCGGTAGTAGGTTC-3′OligoB3: 5′-GAGGCGAGCGGTCAATTTTCCGCGCAAAGAGAAAAGTACGACAAACCCT-3′Deeplexicon 4OligoA4: 5′-/5Phos/GGTCTTCGCTCGGTCTTATTTAGTAGGTTC-3′OligoB4: 5′-GAGGCGAGCGGTCAATTTTAATAAGACCGAGCGAAGACCGACAAACCCT-3′Poreplex 1Oligo A: 5′-/5Phos/CCTCCCCTAAAAACGAGCCGCATTTGCGTAGTAGGTTC-3′Oligo B: 5′-GAGGCGAGCGGTCAATTTTCGCAAATGCGGCTCGTTTTTAGGGGAGG GACAAACCCT-3′Poreplex 2Oligo A: 5′-/5Phos/CCTCGTCGGTTCTAGGCATCGCGTATGCTAGTAGGTTC-3′Oligo B: 5′-GAGGCGAGCGGTCAATTTTGCATACGCGATGCCTAGAACCGACGAGG GACAAACCCT-3′Poreplex 3Oligo A: 5′-/5Phos/CCTCCCACTTTCACACGCACTAACCAGGTAGTAGGTTC-3′Oligo B: 5′-GAGGCGAGCGGTCAATTTTCCTGGTTAGTGCGTGTGAAAGTGGGAGG GACAAACCCT-3′Poreplex 4Oligo A: 5′-/5Phos/CCTCCTTCAGAAGAGGGTCGCTTCTACCTAGTAGGTTC-3′Oligo B: 5′-GAGGCGAGCGGTCAATTTTGGTAGAAGCGACCCTCTTCTGAAGGAGG GACAAACCCT-3′

### 4.11. Oxford Nanopore Library Preparation

Oxford Nanopore sequence-specific direct RNA protocol DSS_9081_v2_revK_14Aug2019 was used in this study. Note that the reverse transcription step to linearise the 28s rRNA was performed as the secondary structure inhibits translocation through the nanopore. For both barcoded and unbarcoded library preparations, 1.4 mM of oligos A and B was annealed 1:1 in 10 mM Tris-HCl pH 7.5 and 50 mM NaCl as described in the direct RNA protocol. For each sample, whether barcoded or not, 2 mg of extracted RNA was split into two 1 mg samples. Each sample proceeded through the Oxford Nanopore protocol as directed, until step 18, and eluted in 10 mL of nuclease-free water. All samples were pooled at step 20 and adjusted with the NEBNext Quick Ligation Reaction Buffer as needed for the volume of reverse-transcribed RNA. The protocol then proceeded as normal.

### 4.12. Oxford Nanopore Sequencing

Sequencing was undertaken on MinION MD106 flow cells, with the SQK-RNA002 kit appropriately selected in Minknow. Guppy version 4.4.1 was used for basecalling. Older runs were re-basecalled with version 4.4.1 for parity.

### 4.13. Demultiplexing

Poreplex [23] was used to demultiplex individual samples using the options: –trim-adapter –barcoding –fast5. For one set of samples, DeePlexiCon [24] was used with default options. Note that any future direct RNA barcoding and demultiplexing tools which work in a similar method to the above tools will likely produce a satisfactory output. 

### 4.14. Quantification of Depurination

Quantification of depurination was achieved using the following workflow:Concatenate all fastqs in fastq_pass per sample;Map concatenated fastqs to NR_003287.4 using: minimap2 -ax splice -uf -k14 ref.fa reads.fq > aln.sam minimap2 [25] version 2.17-r941;Sort and index sam file using Samtools Samtools [26] version 1.9;Run rippore.py to calculate per base nucleotide counts. By default, use base 4605; if using a different 28s rRNA reference to NCBI: NR_003287.4, provide the correct base with -b option. rippore.py -s sample.sam. RIPpore can be found at https://gitlab.com/yryan/rippore, accessed on 6 August 2021.

### 4.15. Statistical Analysis

Summary statistics and Figure 2 were generated in R version 4.0.2 using results from rippore.py with the -o option to create a CSV of nucleotide counts at the depurination site. Significance was determined using the ggsignif package to perform a series of independent *t*-tests, comparing unexposed controls to each exposure condition. 

### 4.16. Raw Signal Analysis

Nanocompore [27] version 1.0.3 was used to identify changes in charge intensity and dwell time at a range of locations on the 28s ribosome. This analysis used NCBI NR_003287.4 as a reference. Nanocompore was also used to generate simulated reads of the 28s ribosome, both wildtype and modified to T4605. The Nanopolish SimReads function was used in its default settings, and no modification options were enabled.

Nanocompore was modified to increase the simulated read dpi, move legends, and increase font sizes for publication purposes.

The references used were:

>WT

AGTACGAGAGGAACCG

>Uracil

AGTACGTGAGGAACCG

>Guanidine

AGTACGGGAGGAACCG

>Cytosine

AGTACGCGAGGAACCG

Tombo [15] version 1.5.1 was used to generate aggregate plots of 50 random reads around the ricin loop, for WT and A4605, depurinated, and T4605 to assess changes in read signal against Nanocompore-simulated signals.

## Figures and Tables

**Figure 1 toxins-14-00470-f001:**
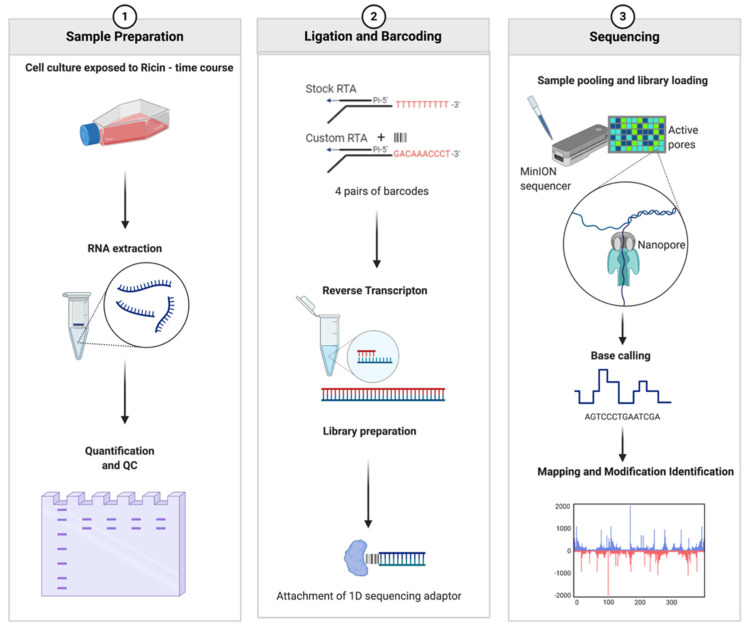
Strategy for measuring the effects of ricin using a direct RNA sequencing approach. A549 cells were exposed to ricin and total RNA extracted from cells and then purified in a continuous procedure. A select portion of rRNA potentially containing the A4605 modification was directly sequenced on a flow cell. Software was developed, RIPpore, that took the output data and quantified the adenine modification.

**Figure 2 toxins-14-00470-f002:**
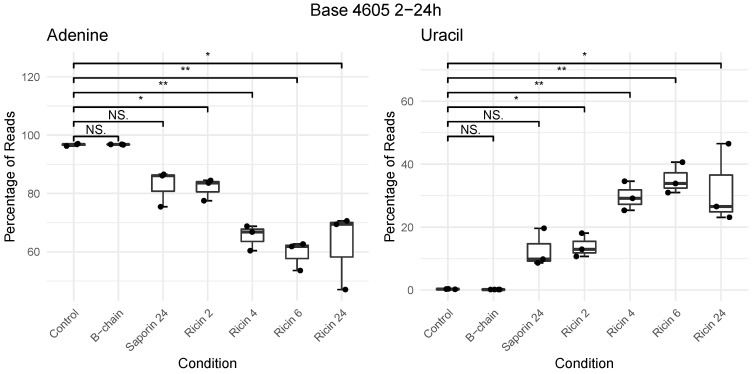
Boxplots showing a time course (2–24 h) of the percentage of nucleotide composition at position A4605 in 28S rRNA from A549 cells following exposure to vehicle control, ricin B-chain, saporin, or pure ricin. Either control, B-chain (6 h), or saporin at 24 h, or ricin at 2, 4, 6, and 24 h post-exposure. Pairwise *t*-tests assessing control to each exposure were used to perform significance testing. NS denotes *p* > 0.05, * *p* < 0.05 and ** *p* < 0.01. See Appendix A for all four nucleotides.

**Figure 3 toxins-14-00470-f003:**
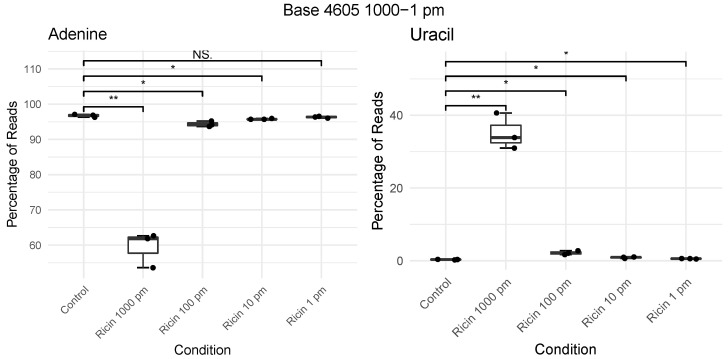
Boxplots showing percentage of nucleotide composition at position A4605 in 28S rRNA from A549 cells exposed to differing concentrations of ricin (1–1000 pm; 6 h). Pairwise *t*-tests assessing control to each exposure were used to perform significance testing. NS denotes *p* > 0.05, * *p* < 0.05 and ** *p* < 0.01. See Appendix A for all four nucleotides.

**Figure 4 toxins-14-00470-f004:**
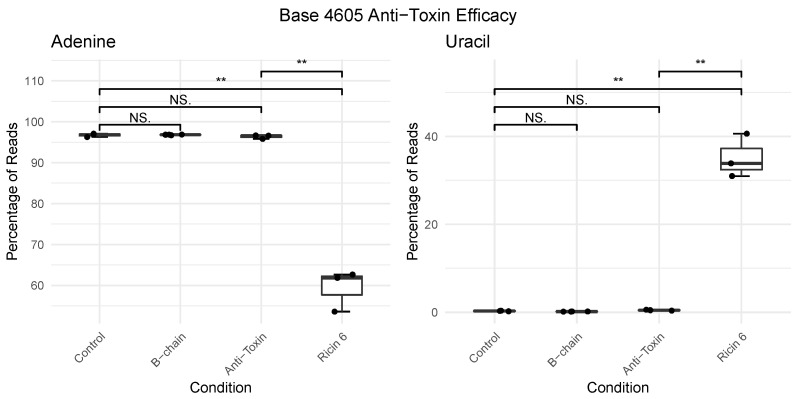
Boxplots showing the percentage of nucleotide composition at position A4605 in 28S rRNA from A549 cells following exposure to vehicle control, ricin B-chain (1 nm, 6 h), ricin (1 nm, 6 h), ricin, and antitoxin (1 nm ricin, neutralising dose antitoxin, 6 h). Pairwise *t*-tests assessing control to each exposure were used to perform significance testing additionally with ricin, and antitoxin against ricin only. NS denotes *p* > 0.05, ** *p* < 0.01. See Appendix A for all four nucleotides.

**Figure 5 toxins-14-00470-f005:**
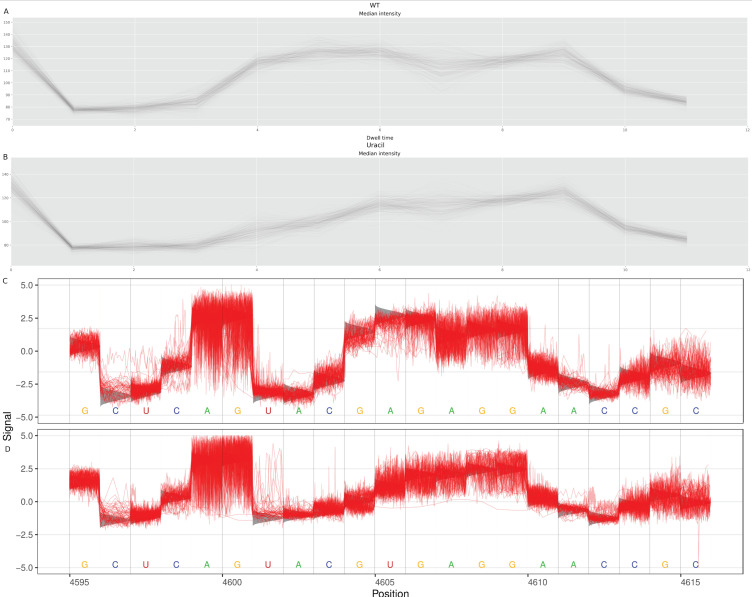
Comparison of Nanocompore median simulated charge intensity with the canonical ricin loop reference sequence (**A**), and sequence with the apurinic site replaced with an uracil (**B**). For (**A**,**B**), the *x*-axis is the reference position, and the *y*-axis is charge intensity (**C**). Tombo output displaying the charge intensity of the ricin loop from control (nonexposed) A549 cells (**D**). Tombo output for the ricin loop from cells exposed to ricin for 6 h, mapped to a reference where the A4605 is altered to U4605. For (**C**,**D**), the *x*-axis is the position of the nucleotide on the 28s RNA and the *y*-axis is a normalised charge intensity of the actual (red) and predicted (grey) sequenced reads.

**Figure 6 toxins-14-00470-f006:**
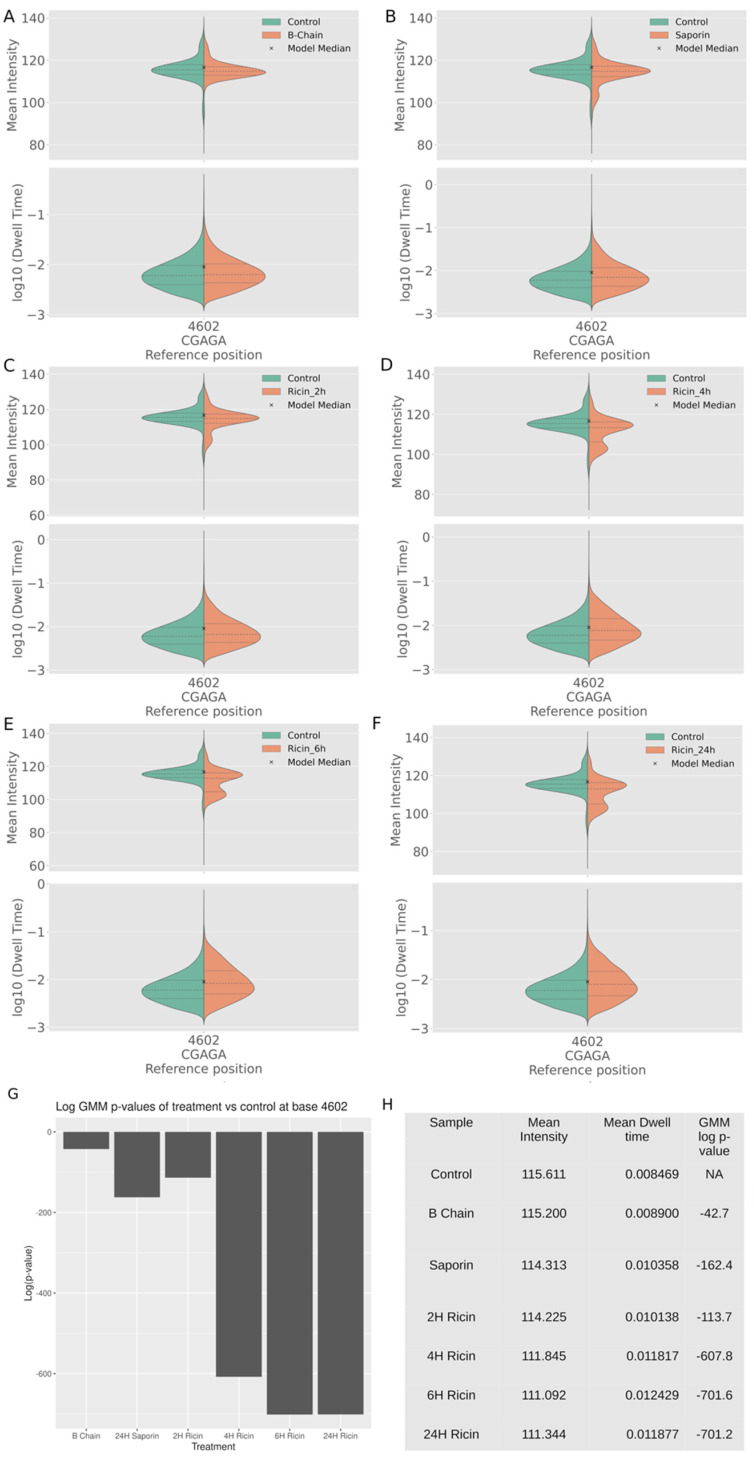
Charge density shown as a violin plot of the charge intensity per read at position 4605 on the 28s RNA determined by Nanocompore analysis of the direct RNA sequencing data (**A**–**F**). Shown are the dwell time and charge intensity at position 4605 from either A549 cells exposed to ricin for 2, 4, 6, and 24 h (**A**–**D**), and ricin B-chain (**E**) and saporin (**F**). The control exposure is shown in green and the relevant experimental exposure in orange. The x indicates the mean, the middle-dashed line the medium, and the outlying dotted lines the interquartile ranges. (**G**) To identify significant differences between the exposures, a pairwise comparison of the logged *p*-values of Gaussian mixture models was used. (**H**) Summary of the mean intensity, dwell time, and the log *p*-value of the difference conditions.

**Figure 7 toxins-14-00470-f007:**
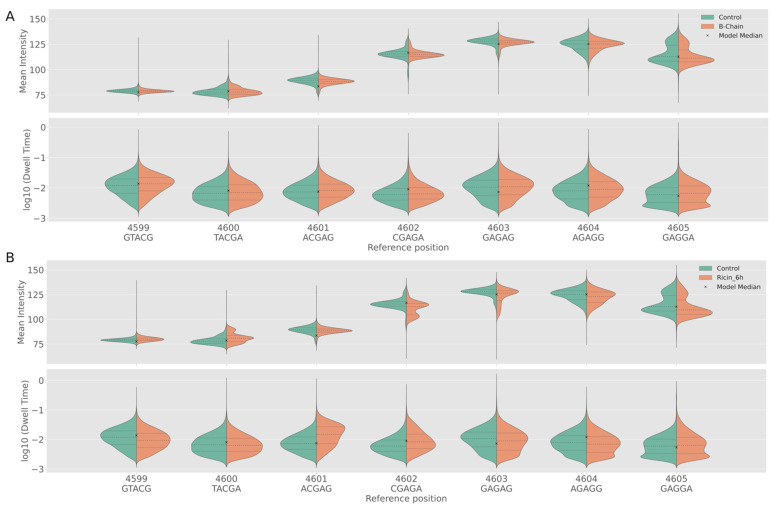
Nanocompore-generated violin plots of charge density and dwell time of: (**A**) Control vs. B-chain and (**B**) Control vs. 6 h ricin exposure. The control exposure is shown in green and the relevant experimental exposure in orange. The x indicates the mean, the middle-dashed line the medium, and the outlying dotted lines the interquartile ranges. B-chain and controls gave similar profiles. For (**A**,**B**), the order of the violin plots represented the sequence around the ricin lop, starting with site 4599 (**left**) and extending to site 4605 (**right**). Ricin exposure caused a decrease in charge at sites 4602 and 4603, but an increase in charge at position 4600. Site 4605 and, to a greater degree, 4601 increased in dwell time in RNA from cells exposed to ricin.

**Table 1 toxins-14-00470-t001:** Table of descriptive statistics of percentage of reads called as uracil by Guppy. Min and max denote lowest and maximum percentages of uracil across each condition, respectively.

Exposure	Min	Max	Median	Mean	SD
Control	0.21	0.34	0.28	0.28	0.06
B-Chain	0.15	0.17	0.16	0.16	0.001
Anti Toxin	0.38	0.60	0.48	0.49	0.11
Saporin	7.55	16.97	8.52	11.02	5.18
Ricin 2 h	9.18	15.34	11.65	12.04	3.13
Ricin 4 h	24.77	29.71	26.48	25.98	3.40
Ricin 6 h	26.54	35.06	30.66	30.76	4.26
Ricin 24 h	19.78	39.64	23.81	27.74	10.50
Ricin 100 pm	1.68	2.75	2.07	2.17	0.54
Ricin 10 pm	0.64	1.04	0.96	0.88	0.21
Ricin 1 pm	0.46	0.61	0.58	0.55	0.08

## Data Availability

Raw Fast5 files available from SRA. BioProject PRJNA752594 (https://www.ncbi.nlm.nih.gov/bioproject/PRJNA752594, accessed on 6 August 2021). RIPpore can be found at https://gitlab.com/yryan/rippore, accessed on 6 August 2021.

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
