# Peer review of "RIPpore: A Novel Host-Derived Method for the Identification of Ricin Intoxication through Oxford Nanopore Direct RNA Sequencing"

_toxins, 2022, doi:10.3390/toxins14070470_

Round 1

Reviewer 1 Report

In this manuscript, the authors use direct RNA sequencing with Oxford Nanopore Technologies to detect modification or 28S rRNA A4605 by ricin toxin. They show that when cells are exposed to ricin, modification at site A4605 is specifically detected by nanopore sequencing. The depurination is interpreted and read as U4605. In a second part, the authors explain how the depurination modification changes charges and local environement and is read as U by Guppy algorithm. The work is intersting and describe a novel assay which high sensitivity for detecting ricin activity.

There are few minor points:

- The authors could compare their assay with other assays precioulsy published (PCR extension qRT-PCR) for detecting depurination at A4605. This would highlight the advantages of their technique is terms of sensibility, efficiency and speed.

- The authors used simulated sequenced data for sequence containing U4605. Why have they not used an RNA oligo containing U4605?

A direct measurement of an RNA molecule containing really U4605 would be interesting to find specific differences from depurination to A/U substitution in RNA. If this information exists, it would be useful for training guppy to distinguish depurination from A to U substitution.

Author Response

Dear Reviewer 1.

Many thanks for your feedback and taking the time to review our article. Please see the following responses to your feedback which I have added to the discussion, but included here beneath each point for clarity.

- The authors could compare their assay with other assays precioulsy published (PCR extension qRT-PCR) for detecting depurination at A4605. This would highlight the advantages of their technique is terms of sensibility, efficiency and speed.

We thank the reviewer for requesting further clarification on this point and have added this section into the discussion which we believe addresses this matter.

            “However, we believe our method may have more accurate results as it relies on a fully processed 28s rRNA in order for the oligo to bind to the final 10 3’ bases, whereas PCR methodologies will likely count any immature 28s rRNA transcripts present.”

- The authors used simulated sequenced data for sequence containing U4605. Why have they not used an RNA oligo containing U4605?

We thank the reviewer for their contribution and note that they raise very interesting point. We had considered this as an approach however there was considerable (technical) risk in adopting the analysis of an oligo in this way given that ONT typically struggles with reads shorter than 150bp. As this wasn’t the specific question being asked we used simulated data in this instance. Nonetheless we have added a short extra section to the discussion to clarify this specific point which we hope will now satisfy the reviewer.

"Simulated data was used instead of generation of ssRNA oligonucleotides containing U4605, as the synthesis process is limited to a maximum size of ~80 bp whereas ONT sequencing has historically struggled with reads shorter than 150 bp. It is only recently that a software update has been made available that can accurately identify and basecall shorter read lengths to a minimum of 20bp in length (16)."

-A direct measurement of an RNA molecule containing really U4605 would be interesting to find specific differences from depurination to A/U substitution in RNA. If this information exists, it would be useful for training guppy to distinguish depurination from A to U substitution.

We thank the reviewer for their obervation and note that this point merited discussion. We hope the following addition to the discussion satisfies this.

"The issue of read lengths and fact that bonito (i.e. the ONT tool used to train basecaller models for guppy), are not designed to work on bases outside of ATCGU makes it difficult to train a basecaller that is able to detect depurination. In the future, if the synthesis or read length of oligos containing uracil at base 4605 can be managed, it may be theoretically possible to train a basecaller that can identify the depurination of base 4605 and that could differentiate an abasic site from U4605. This would require a tool to specifically replace U at depurinated bases in the basecalled data with an appropriate symbol (D for depurination, or R for ribose backbone) as part of the training dataset and modification of Bonito (currently used by ONT to train basecalling models) to handle additional bases outside of ATCGU which is not available at present or planned, to the best of our knowledge."

Reviewer 2 Report

GENERAL COMENT

In this manuscript, the authors report “RIPpore; a new host-derived method for the identification of ricin poisoning via Oxford Nanopore direct RNA sequencing”

The manuscript deals with an interesting topic and the experiments are well designed and properly carried out.  Ricin and saporin are N-glycosylases that catalyze the depurination of an adenine base in the SRL of 28S rRNA, leading to inhibition of protein translation and cell death. Furthermore, both RIPS have been widely used in the preparation of immunotoxins to attack tumor cells, and it has been shown that these RIPs alone, or as a toxic component of the immunotoxin induce apoptotic cell death. The main contribution of the authors of this manuscript is the fact that they have developed a new methodology, based on the use of ONT dRNA sequencing to specifically detect and quantify the adenine deletion caused by RIPs. The results obtained by the authors have demonstrated the usefulness of the technique in lung epithelial cells (A549) cultures exposed to ricin and saporin.

On this basis, the article can be accepted for publication with minor changes.

MINOR CHANGES

Introduction

Line 37: Please, modify this sentence: ….The B chain is a lectin that  binds…….

Lines 40-41.: Please, correct this paragraph and add an appropriate reference on saporin cell endocytosis.

It has been described in the literature that Saporin can enter intoxicated cells through different mechanisms depending on the cell type. Initially, it has been proposed that endocytosis may occur by passive mechanisms such as fluid phase pinocytosis, however, in some cell types Saporin-S6 uptake has been observed to occur by receptor-mediated endocytosis via the macroglobulin receptor α2 (also called LRP). Electron microscopy experiments have indicated that endocytosis of saporin-S6 by HeLa cells occurs primarily via uncoated vesicles.

 (Reference: Letizia Polito *, Massimo Bortolotti, Daniele Mercatelli, Maria Giulia Battelli and Andrea Bolognesi . Saporin-S6: A Useful Tool in Cancer Therapy. Toxins 2013, 5, 1698-1722; doi:10.3390/toxins5101698 toxins ISSN 2072-6651).

On the other hand, it is clear that the use of transfection reagents increases saporin entry into cells (Reference 19).

Line 59:  Please, change “base caller” by “basecaller”

Results

Line 131: Please, change “28s ribosome” by “28S rRNA”

Fig. 6 and 7: increase the font size of the title of the axes

Discussion

Line 294: Please, remove “transfected” in this sentence ………cells exposed to ricin and saporin……….  

Line 313: Please, change “28s ribosome” by “28S rRNA”

Lines 269-270 - Legend of Fig. 7: Please, check this sentence:….. Ricin exposure caused a decrease in charge at sites 4602 and 4603, but an increase in charge at position 4603.

¿ should it be 4600?……

Author Response

Dear Reviewer 2,

Many thanks for your feedback and taking the time to review our article. Please see the following responses/changes to your feedback. We greatly appreciate your comments and hope our changes have successfully rectified your concerns.

Introduction

Line 37: Please, modify this sentence: ….The B chain is a lectin that  binds…….

            Changed to: Ricin-B chain binds to cell surface galactosides and facilitates entry whilst the A chain catalyses depurination, classifying ricin as a type II RIP.

Lines 40-41.: Please, correct this paragraph and add an appropriate reference on saporin cell endocytosis.

It has been described in the literature that Saporin can enter intoxicated cells through different mechanisms depending on the cell type. Initially, it has been proposed that endocytosis may occur by passive mechanisms such as fluid phase pinocytosis, however, in some cell types Saporin-S6 uptake has been observed to occur by receptor-mediated endocytosis via the macroglobulin receptor α2 (also called LRP). Electron microscopy experiments have indicated that endocytosis of saporin-S6 by HeLa cells occurs primarily via uncoated vesicles.

 (Reference: Letizia Polito *, Massimo Bortolotti, Daniele Mercatelli, Maria Giulia Battelli and Andrea Bolognesi . Saporin-S6: A Useful Tool in Cancer Therapy. Toxins 2013, 5, 1698-1722; doi:10.3390/toxins5101698 toxins ISSN 2072-6651).

On the other hand, it is clear that the use of transfection reagents increases saporin entry into cells (Reference 19).

            Altered to:

"Related type I RIPs, such as saporin, lack this B chain, therefore typically require the use of an artificial cell entry mechanism (i.e., transfection reagent or antibody conjugation) to enter a cell and cause toxicity. Saporin can dependent on cell type, enter cells via endocytosis of uncoated vesicles (5) or binding to the macroglobulin receptor α2 (6)."

Line 59:  Please, change “base caller” by “basecaller”

            Removed space

Results

Line 131: Please, change “28s ribosome” by “28S rRNA”

Changed

Fig. 6 and 7: increase the font size of the title of the axes

            Font sizes increased

Discussion

Line 294: Please, remove “transfected” in this sentence ………cells exposed to ricin and saporin………. 

            Changed to: RNA obtained from cells exposed to ricin and saporin internalised via transfection reagents

Line 313: Please, change “28s ribosome” by “28S rRNA”

Changed

Lines 269-270 - Legend of Fig. 7: Please, check this sentence:….. Ricin exposure caused a decrease in charge at sites 4602 and 4603, but an increase in charge at position 4603.

¿ should it be 4600?……

            Corrected to 4600